# Estimation of Knee Joint Angle Using Textile Capacitive Sensor and Artificial Neural Network Implementing with Three Shoe Types at Two Gait Speeds: A Preliminary Investigation

**DOI:** 10.3390/s21165484

**Published:** 2021-08-14

**Authors:** Vantha Chhoeum, Young Kim, Se-Dong Min

**Affiliations:** 1Department of Software Convergence, Soonchunhyang University, Asan-si 31538, Korea; vantha.chhoeum@gmail.com; 2Institute of Wellness Convergence Technology, Soonchunhyang University, Asan-si 31538, Korea; ykim02@sch.ac.kr; 3Department of Medical IT Engineering, Soonchunhyang University, Asan-si 31538, Korea

**Keywords:** artificial neural network, gait speed, knee joint angle, foot plantar pressure, shoe type, textile capacitive sensor

## Abstract

The lower limb joints might be affected by different shoe types and gait speeds. Monitoring joint angles might require skill and proper technique to obtain accurate data for analysis. We aimed to estimate the knee joint angle using a textile capacitive sensor and artificial neural network (ANN) implementing with three shoe types at two gait speeds. We developed a textile capacitive sensor with a simple structure design and less costly placing in insole shoes to measure the foot plantar pressure for building the deep learning models. The smartphone was used to video during walking at each condition, and Kinovea was applied to calibrate the knee joint angle. Six ANN models were created; three shoe-based ANN models, two speed-based ANN models, and one ANN model that used datasets from all experiment conditions to build a model. All ANN models at comfortable and fast gait provided a high correlation efficiency (0.75 to 0.97) with a mean relative error lower than 15% implement for three testing shoes. And compare the ANN with A convolution neural network contributes a similar result in predict the knee joint angle. A textile capacitive sensor is reliable for measuring foot plantar pressure, which could be used with the ANN algorithm to predict the knee joint angle even using high heel shoes.

## 1. Introduction

The lower limb joint angles of optimal performance may be affected by different gait speeds and shoe types. Modification of gait speed may change spatiotemporal parameters, joint kinematics, joint kinetics, and ground reaction forces [1]. In addition, the fluctuating function of the metatarsophalangeal joint and other mechanics of lower limb joints may be affected by different shoe outsole structures [2]. The lower limb joint in humans, especially women, could be influenced by the type of shoe worn, which is mainly due to increasing shoe heel height [3,4]. The increasing knee range of motion was found in fast gait speed compared to comfortable gait speed [5] and the knee flexion also appears to increase when increasing the height heel [3]. Observations of the effects of altering human gait not only benefit clinical and medical studies but are also crucial to the fields of sport, rehabilitation, training, and robotic research [6]. Monitoring human joint angles requires skill and proper technique to obtain accurate and reliable data. Several systems and technologies have few limitations that can be used to measure joint angles, including motion capture systems, clinical tools, and sensor application.

Imaging and video-based tracking systems and motion capture systems provide good results when measuring joint angles due to multiple camera usage but these systems require a large space and professional technical setup, which is a costly undertaking [7,8]. The universal goniometer is the clinical tool defined as the gold standard for assessing joint angles [9] but requires two hands to manipulate the instrument, which might cause errors during the measuring process [10]. Moreover, joint angles can also be obtained from application of sensors such as optical fibers [11,12], optical-based goniometers [13], textile-based conductive wires [14], textile-based flex fabric [15], textile-based strain fabric [16], smartphone sensors [17], gyroscopes [18], and inertial measurement units (IMU) [19]. Measuring joint angles with those sensor technologies is still under a few limitations related to sensitivity, low accuracy, nonlinearity, complex algorithms, and they require accurate sensor alignment [20]. Recently, advancements in new technologies that use a machine [21,22] and deep learning approach [23,24] with sensor technologies have contributed to developing various algorithms that provide accurate and effective predictions for lower limb joint angles giving reliable results for further analyses. Moreover, to make the system more convenient and easier to implement, the user avoids finding the optimal sensor placement, so the current method of joint angle assessment could conduct through foot plantar pressure and a deep learning algorithm.

The foot plantar pressure in the form of ground reaction force has been applied to deep learning of an artificial neural network (ANN) in estimating ankle joint angle [25]. Sivakumar et al. (2019) indicated that the ground reaction force could have the capability to assess ankle joint angle relied on the ANN approach. Moreover, integrating foot pressure data with multi-source data, including surface electromyography (sEMG), hip joint angle, provides high accuracy in building a deep learning model in estimating lower limb joint angles [23]. The model provided a better correlation efficiency and short calculation times compared to a backpropagation neural network [23]. The foot plantar pressure has been shown the capability, although it can stand alone or integrate with multi-sources signals, to serve as input data for deep learning to assess the joint angles. Therefore, the foot plantar pressure plays an essential role in building a deep learning model, and the most common system used to measure plantar pressure are the platform system and the in-shoe system [26,27].

The plantar platform system includes a rigid flat platform, while the in-shoe system is flexible and embedded in shoes, which is a more suitable application for various shoe and heel heights [26,27]. The in-shoe system could use various types of pressure sensors placed to the insole that can be applied with variation shoe designs. The most common pressure sensors are the capacitive sensors, resistive sensors, piezoelectric sensors, and piezoresistive sensors [26]. Moreover, recently many researchers have developed various pressure sensors to measure foot plantar pressure with high performance and effectiveness, which can apply in many applications. Zhang et al., (2019) proposed the development of a textile capacitive sensor that could use to measure foot plantar pressure. The textile capacitive sensor provides good performance in signal response, hysteresis, and stability with a less complex fabrication process resulting in a low cost [28]. Due to those advantages of a textile capacitive sensor and it also can apply with variation shoe design, we propose the development of a textile capacitive sensor with simple structure design, wireless, and less costly placing in insole shoes to measure the foot plantar pressure for building deep learning model.

The deep learning approach can use input data directly, so it omitted the process of feature extraction that might take time and costly operation. Recently, the deep learning approach of ANN [25] and convolutional neural network (CNN) [29] have been implemented with various data source to estimate joint angles. The CNN approach is required to employ the convolute technique to the input data, which needs to consider few factors, including the number of filters, filter size, and pooling method. This process may cause information leakage affecting model training. So, obtaining joint angles using the CNN approach is challenging due to the convolute technique. The ANN approach may consider as a simple neural network consisting of three or more layers; input layer, hidden layer(s), and output layer. The data pass through various nodes till to output node in one direction. Moreover, the potential of ANNs could generalize to a wider population, and estimation through ANNs does not require anthropometric information [30]. Therefore, the main aims of this study were to (1) observed the textile capacitive sensor’s characteristics as a pressure sensor that generated input data to create an ANN model, and (2) evaluated the ANN algorithm depending on foot plantar pressure only to predict knee joint angle when using three shoe types at two gait speeds.

## 2. Materials and Methods

### 2.1. Design of Foot Plantar Pressure System

#### 2.1.1. Textile Capacitive Sensor

Construction of the capacitive sensor using three electrodes increases the capacitance in equivalence compare to two electrodes. And adding a third plate produces a more stable ratiometric output signal [31]. Moreover, placing the ground electrode on the top and bottom could minimize friction noise and proximity effects caused by the shoe. Figure 1a indicates the design structure of the textile capacitive sensor and Figure 1b shows the equivalence structure of each layer. The textile capacitive sensor was developed to measure the foot plantar pressure using three shoe types at two gait speeds. Under pressure and the inclination of the high heel shoe may cause slippering in between sensor layers, so this factor needs to minimize through the slipping resistance material. The nonwoven dielectric material has a rough surface that could function as slip-resistant, inducing less slippering between sensor layers. Our sensors were made by electrode plates using the conductive textile of W-209-PCN (A-jin Electron, Gangseo, Busan, Korea) and non-woven fabric dielectric has a thickness of 1.00 × 10^−3^ m. 

The capacitance of the sensor can also calculate using a formula as express below:(1)$$C=\varepsilon_{r}\varepsilon_{o}\frac{A}{d},$$
where $\varepsilon_{r}$ is the relative permittivity of the dielectric material of non-conductive rubber, $\varepsilon_{0}$ is the vacuum permittivity, A is the area of the electrode, d is the distance between two electrodes.

Figure 2a shows a schematic of hardware design to measure the capacitance of the textile capacitive sensor. And Figure 2b shows the detail and connection of hardware design based on the development by Wang et al., (2018) [32]. The MPR121QR2 was used to convert the analog to a digital signal. MPR121QR2 can measure sensor frequency up to 400 kHz, and capacitance range from 10 pF to 2000 pF with a resolution of 0.01 pF. The development of STMicroelectronics (Geneva, Switzerland) served as a microcontroller unit used to measure the capacitance. The data were sampled with a sampling frequency of 100 Hz and sent to a computer through Bluetooth communication.

#### 2.1.2. Performance Testing

The upper part of our sensor design was a textile conductive electrode that serves as the ground electrode (Figure 1). In our experiment protocol, the participants were required to wear socks that prevent direct contact with the sensor. Therefore, in performance testing of our development sensor, we covered the sensor’s surface with the same socks to avoid direct contact with the sensor and metallic weight. 

Before we test the performance of our development textile sensor, we defined the limits of the maximum pressure that could apply to our sensor. In our study, the three shoe types have the same size of 23.5 cm with a minimum ball width of 8.0 cm, so the minimum stress area of the insole (23.5 cm × 8.0 cm = 188.0 cm^2^) is 188.0 cm^2^. At mid-stance, one foot fully supports the body weight while the other foot lifts the ground. We defined a subject’s bodyweight as up to 800 N (80 kg × 9.81 m/s^2^). Therefore, the pressure while the foot fully supports the bodyweight based on formula (P = F/S, where F is the bodyweight and S is the stress area of the foot) is 41.74 kPa. Typically, the total stress area of the foot in contact may be less than we defined due to the foot of participant contact to the insole. When the stress area of contact becomes small, so the pressure gets high. The different pressure ranks maybe provide the various trends of sensor performance. To avoid lacking information for generating the trend of changing capacitance verse pressure, we investigated the characteristic of our sensor with pressure higher than define. 

We tested the textile capacitive sensor to observe the relationship of changing capacitance under applied mass from 0 to 10 kg of stainless-steel calibration weight. The mass was incremented by 0.50 kg, which was used on the developed sensor five times, so the capacitance was noted five times. The LCR meter (LCR-8110G, GWInstek, New Taipei City, Taiwan) was used to measure the capacitance of our sensor correspond to applying weight. The performance testing was done on a textile capacitive sensor with a dimension of 0.03 m, 0.03 m, and 1.00 × 10^−3^ m for height, width, and thickness. The sensor was composited with the same sensor structure in Figure 1b. According to the pressure formula (P = F/A; F is the weight of stainless-steel, A is the sensor’s area), it can calculate that the pressure that applies to the sensor is closed to 110 kPa. Figure 3 depicts the performance testing setup for our sensor.

### 2.2. Experimental Setup and Protocol

In this paper, the method proposed to evaluate the artificial neural network (ANN) algorithm relies on foot plantar pressure measures from three shoe types under two gait speeds, making it possible to predict the knee joint angle. Three shoe types were selected for this study, flat (S1), sneaker (S2), and stiletto heel (S3). The two gait speeds of comfortable gait (C) and fast gait (F) were also chosen. Figure 4a shows the lateral side of the three shoe types used in our study. The foot plantar pressure was collected using the development of the textile capacitive sensor. We designed a textile capacitive sensor with a dimension of 0.01 m × 0.01 m. Ten textile capacitive sensors were set up in each insole of three testing shoes. To achieve all plantar pressure data on foot during walking, we arranged the sensor location into three main foot areas; forefoot, midfoot, and rearfoot. Figure 4b shows the sensors set up in the insole to collect the foot plantar pressure of three main foot areas. Table 1 describes the three shoe types characteristics. Figure 5 shows the system setup, indicating sensor layers and hardware design to receive and transmit data.

The kinematic data of the knee joint was measured throughout the experiment. The lateral side of the hip, knee, and ankle joint was marked with a reflective marker, and a smartphone was used to video record the walking of each participant. A tripod held the smartphone (camera). It was set to a height of 1.0 m from the ground and positioned 2.5 m away from the treadmill center [33].

Seven young, healthy women participated in this study. Their average age, height, weight, and body mass index (BMI) were 20.7 ± 0.8 years, 1.58 ± 0.04 m, 50.40 ± 2.20 kg, and 20.30 ± 2.20 kg/m^2^, respectively. The inclusion criteria were any healthy women who have experience of walking in high heels and have used a treadmill for running before. The women who had any leg injury history, surgery in the lower extremity, musculoskeletal diseases, pain in the foot area, and flat foot were excluded from our study.

First, we informed the main objective in our study and clearly explained the experimental protocol to all participants. The order of wearing 3 test shoes and choosing the two gait speeds were all randomized by computer software. Each participant was instructed to start walking on the treadmill to get familiar with the environment and, at the same time, to self-select the comfortable and fast gait speed. The participants were required to wear socks that prevent direct contact with the sensor. A total of 6 experimental conditions were performed by each participant with three shoe types, S1, S2, and S3, and two gait speeds, C and F. Participants were required to walk for 2 min on a treadmill for each experimental condition and had 5-min breaks in between. Our experiment protocol was approved by the ethics committee of the Institutional Review Board of Soonchunhyang University (No.1040875-202012-BM-091).

### 2.3. Data Processing

The Kinovea 0.8.27 (Open and Free Software Foundation, Inc., Boston, MA, USA) is the motion capture software used to calibrate the knee joint angle from the video recording of each experiment condition. After calibration, the software can save the knee joint angle data as the excel field for further processing. MATLAB R2015b (MathWorks, Inc., Natick, MA, USA) is a software containing many libraries that can be used for signal processing. MATLAB R2015b was used to process raw foot plantar pressure data. It was also used to extract 30 gait cycles (GCs) data from the knee joint angle and foot plantar pressure then normalize each gait cycle data to 100 data points. The EXCEL is a powerful data visualization and analysis tool used to generate the trend of capacitance variation regarding varying pressure. Python 3.7 is free software containing many libraries that can create deep learning models [34]. 

In this paper, a development of textile capacitive sensor was used to collect the foot plantar pressure. The textile capacitive sensor serves as a pressure sensor consisting of ten sensors mounted on the left insole and can measure the data at a sampling rate of 100 Hz. These measured data were transmitted to the computer through a Bluetooth application and save as an excel file for further processing. Our study used the knee joint angle data as the reference. The smartphone was used to record the video data with a 30 Hz frame rate, and the video data were transferred to a computer by a USB cable that can operate with motion capture software to extract knee joint angle.

#### 2.3.1. Foot Plantar Pressure

During the experiment, the sensor’s raw foot pressure data was transferred from the microcontroller through Bluetooth communication to a personal computer. The software-based on C# language displays the raw data then saves data to an excel file for further processing. The 2-min raw data of foot plantar pressure was proccing in MATLAB 2015b. Figure 6 shows the workflow of processing the raw pressure data to obtain 30 GCs data from each subject and experiment condition used as an input to train the ANN model. 

All subjects were asked to walk on the treadmill for 2 min for each experimental condition. To obtain stable and useful data for building the model, we trimmed 1-min data; cut out some data from the beginning and ending of each trial out of 2 min data. The foot plantar pressure was calculated regarding the trend of changing capacitance obtained from the performance test of the textile capacitive sensor. Each insole consisted of ten sensors, so we calculated the summation of the pressure of all sensors, which was defined as the total plantar pressure that could generate during walking.

The raw plantar pressure data contain the noise made by the treadmill and friction between sensor and insole, so 10 points of moving average filter were applied. Normally, the frequency of the foot plantar pressure signal in humans was below 15 Hz [35], so the Butterworth low pass filter with 4th order and a cut-off frequency of 15 Hz was applied to remove the unwanted signal.

Our work intended to evaluate a deep learning approach’s ability to predict the knee joint angle throughout GC data. We defined a GC data contain 100 input features which used to build the model. Moreover, deep learning model require large dataset for training to get the good performance model. Therefore, for each subject and experimental condition, 30 GCs of foot plantar pressure were extracted as input for building models.

Typically, the foot plantar pressure appears to higher than zero value (*p* > 0 kPa) when the foot contacts the ground, so no contact foot plantar pressure may go to zero. We defined the first zero value of plantar pressure appears at the toe-off gait event. We created the Zero-To-Zero detection algorithm to segment the GC on mid-1-min data. The Zero-To-Zero detection defined a GC data at toe-off to next toe-off gait event. So, we extracted 30 GCs regarding Zero-To-Zero segmentations on mid-1-min data. Due to a different subject and experimental condition, the length of a GS was unequal. To obtain 100 data of a GC, we normalized each GC data to 100% data with the interpolation technique of interp1 in the 1D function on MATLAB. Then we reconstruct the GC at toe-off to heel strike gait event based on gait interval percentage. 

#### 2.3.2. Knee Joint Angle

The 2-min video data were calibrated through the motion capture software of Kinovea. The video was trimmed to 1 min by cut out some data from the beginning and ending. In Kinovea, the marker on the hip, knee, and ankle were connected using an angle tool. Figure 7 demonstrates the sample image on how to measure the knee joint angle. From the video of each experimental condition, the angle tool was assigned to start the track at the heel strike angle and end at the next heel strike, and the data was export as an excel file. For each subject and experimental condition, 30 GCs of knee joint angle data were extracted as the reference data for the ANN model. Due to the unequal length of each GC data, the knee joint angle was normalized using the same technique as plantar pressure data. Balancing input and reference data may make the machine easy to learn during training, contributing to the high accuracy of prediction.

### 2.4. Artificial Neural Network (ANN)

The workflow of data processing of prediction on knee joint angle based on plantar pressure data with the ANN algorithm is shown in Figure 8. We used the data of five subjects for building the ANN model and two subjects for testing the model performance. The data set is normalized to enhance the training and convergence speed of neural networks by reducing the wide range of amplitude differences [23]. Equation (2) is applied to normalize the data set.
(2)$$x_{\mathrm{norm}}=\frac{x-x_{\min}}{x_{\max}-x_{\min}},\;$$

The $x_{\mathrm{norm}}$ results from input normalize in the range of 0 and 1, x is the input data, $x_{\min}$ is the minimum value of x data, and $x_{\max}$ is the maximum value of x data.

#### 2.4.1. Propose ANN Models 

In our study, there is a total of six experimental conditions; three shoe types and two gait speeds. We extracted 30 GCs from each experimental condition, and GC data consisted of 100 data points. The number of the GC and dataset conditions for training and validation used to develop a specific mode are shown in Table 2. In our research, a total of six ANN models was proposed, three shoe-based ANN models, two speed-based ANN models, and all condition ANN model. The shoe-based ANN model was defined when the ANN model relied on a foot plantar pressure dataset collected from a single shoe at comfortable and fast gait speed. The speed-based ANN mode was set when the ANN model used a foot plantar pressure dataset collected from three shoe types at each gait speed. We investigated each model’s performance to see which parameters (shoe-based or speed-based) could have more capable of producing the model in deep learning that provides better results to estimate the knee joint angle. The six ANN models are listed below which are defined based on using a combination of datasets:M1 is the ANN model using the dataset of shoe type S1 at comfortable and fast gait speed to train the model.M2 is the ANN model using the dataset of shoe type S2 at comfortable and fast gait speed to train the model.M3 is the ANN model using the dataset of shoe type S3 at comfortable and fast gait speed to train the model.M4 is the ANN model using the dataset of three shoe types (S1, S2, and S3) at comfortable gait speed to train the model.M5 is the ANN model using the dataset of three shoe types (S1, S2, and S3) at fast gait speed to train the model.M6 is the ANN model using the dataset of three shoe types (S1, S2, and S3) at comfortable and fast gait speed to train the model.

#### 2.4.2. ANN Algorithm

In general, an ANN consists of three or more layers, including the input layer, hidden layer(s), and output layer. Our paper designed an ANN that can utilize foot plantar pressure data (P) in estimating knee joint angle (θ). Our study used four layers of ANN (input-hidden 1-hidden 2-output) contain (100-250-150-100) nodes to form each model of predicting the knee joint angle. Multiple hidden layers may make the system more flexible and more powerful. To reduce the complexity and introduce the non-linearity of data, three activation functions, Sigmoid, Relu, and Elu, have been chosen. Figure 9 depicts the detailed elements in the block diagram of ANN layers for our study. Each layer consists of the node number (i, j, k, m) and is fully connected by weight. 

During training, each neuron of hidden layer 1 ($h_{j}$) was fed by summation of the linear combination of inputs ($P_{i}w_{\mathrm{ji}}$) and bias ($b_{j}^{\prime }$) for sample n, which is presented in Equation (3). The apostrophe (′) and (″) are defined to identify hidden layer 1 and hidden layer 2, respectively. And each activated neuron of hidden layer 1 ($y_{1j}$) was obtained from applying Sigmoid activation function to $h_{j}$, Equation (4).
(3)$$h_{j}\left(n\right)=\sum_{i=1}^{N^{l}}P_{i}\left(n\right)w_{\mathrm{ji}}+b_{j}^{\prime },$$
(4)$$y_{1,j}\left(n\right)=\sigma \left(h_{j}\left(n\right)\right)=\frac{1}{1-e^{h_{j}\left(n\right)}},$$
where $P_{i}$ is the ith input of sample n, $w_{\mathrm{ji}}$ is defined as the network weight between input ith and hidden layer 1 jth, $b_{j}^{\prime }$ is the bias at hidden layer 1, and $N^{l}$ is the total number of input nodes.

For each neuron of hidden layer 2 ($z_{k}$) was fed by summation of ($y_{1,j}w_{\mathrm{kj}}$) and bias ($b_{k}^{\prime \prime }$) for sample n, which is shown in Equation (5). And each activated neuron of hidden layer 2 ($y_{2,k}$) was obtained by using Relu activation function to $z_{k}$, Equation (6).
(5)$$z_{k}\left(n\right)=\sum_{j=1}^{N^{h}}y_{1,j}\left(n\right)w_{\mathrm{kj}}+b_{k}^{\prime \prime },$$
(6)$$y_{2,k}\left(n\right)=\mathrm{RELU}\left(z_{k}\left(n\right)\right)=\;\max\;\left(0,{\;z}_{k}\left(n\right)\right),$$
where $w_{\mathrm{kj}}$ is defined as the network weight between hidden layer 1 jth and hidden layer 2 kth, $b_{k}^{\prime \prime }$ is the bias at hidden layer 2, and $N^{h}$ is the total number of nodes for hidden layer 1. 

For each neuron of output layer ($y_{3,m}$) was fed by summation of ($y_{2,k}w_{\mathrm{mk}}$) and bias ($b_{m}$) for sample n, which is illustrated in Equation (7). And each neuron of output layer ($\theta_{m}$) was activated by applying Elu activation function to $y_{3,m}$, Equation (8).
(7)$$y_{3,m}\left(n\right)=\sum_{k=1}^{N^{z}}y_{2,k}\left(n\right)w_{\mathrm{mk}}+b_{m},$$
(8)$$\theta_{m}\left(n\right)=\mathrm{ELU}\left(y_{3,m}\left(n\right)\right),\{\begin{array}{ll} \theta_{m}\left(n\right)=y_{3,m} & y_{3,m}\geq 0\\ \theta_{m}\left(n\right)=\alpha \left(e^{y_{3,m}}-1\right) & y_{3,m}<0 \end{array}$$
where $w_{\mathrm{mk}}$ is defined as the network weight between hidden layer 2 kth and output layer mth, $b_{m}$ is the bias at output layer, and $N^{z}$ is the total number of nodes for hidden layer 2.

The loss during training cannot avoid, but we can minimize the loss by introducing the optimizer to form generalization. The Root Mean Square Propagation (RMSpro) optimizer with a learning rate of 0.007 was chosen in our study. The RMSpro updated the weights and biases during the iteration of each epoch. The RMSpro is keeping the moving average of the squared gradients for each weight and bias.

During training, the training dataset were validated by validation data to improve the learning and follow checking the fitting situation for each epoch. All the deep learning approach was used leave one subject out cross-validation to validation model during training.

The result of prediction has to de-normalized to obtain the knee joint angle. While training and validation generally increase the goodness-of-fit, the signal may be distorted. Then we executed the digital low pass filter to signal after de-normalize, which could maintain the same frequency between input and prediction output.

#### 2.4.3. Accuracy Analysis 

The root mean squared error (RMSE) is used to measure the difference between the knee joint angle prediction and the actual value. The RMSE perform over data sample to estimate the error between prediction and actual, and the equation to perform its error illustrated in Equation (9).
(9)$$\mathrm{RMSE}=\sqrt{\frac{1}{N}\sum_{i=1}^{N}{\left(x_{i}-y_{i}\right)}^{2}},$$
where N is the number of samples, $x_{i}$ is the perdition and $y_{i}$ is the actual output. 

The mean relative error (MRE) performs to see the goodness of the model in the fitting. The equation (10) is used to calculate the MRE of each model.
(10)$$\mathrm{MRE}=\frac{1}{N}\sum_{i=1}^{N}\left|\left(x_{i}-y_{i}\right)/x_{i}\right|,$$

Correlation efficiency (R-value) is used to identify the statistical relationship between prediction output and actual output. The formula is shown in Equation (11).
(11)$$R=\sqrt{1-\frac{\sum_{i=1}^{N}{\left(y_{i}-x_{i}\right)}^{2}}{\sum_{i=1}^{N}{\left(y_{i}-\overset{-}{y}\right)}^{2}}},$$
where N is the number of samples, $\overset{-}{y}$ is the mean of actual knee joint angle, $x_{i}$ is the perdition, and $y_{i}$ is the actual output. 

#### 2.4.4. Knee Flexion 

We also observed each model’s performance by looking at knee flexion patterns. The knee flexion was achieved by substrate 180 degrees to an output of knee joint angle prediction, obtained for ANN models. The gait phases of the stance phase (SP) and swing phase (SwP) are defined as the silence phase in the GC. The percentage of each gait phase was defined as (0–60) %, (60–100) % for SP and SwP, respectively.

### 2.5. Convolution Neural Network (CNN)

A convolution neural network (CNN) is also a state-of-the-art model as ANN, so we compared our ANN with the CNN approach to see the system’s flexibility. In our work, the CNN model utilizes 32 filters and 64 filters for the Conv1D_1 layer and Conv1D_2 layer, respectively. Each filter size consists of 3 × 3 and 2 × 2 max-pooling in all convolutions. We employed dropout with a probability level of 50% max-pooling in all convolution layers. We transformed the output of Conv1D_1, Conv1D_2, Dense through the activation function of Sigmoid, Relu, and Elu, respectively. So, the output layer is fed by applying a linear activation function to the Dense layer’s summation output. For training parameters, batch sizes of 32, 50 epochs, and RMSpro optimizer with the learning of 0.007 were set. All the deep learning approach was used leave one subject out cross-validation to validation model during training. All CNN models were trained by using the same combination of the dataset as used in ANN models.

## 3. Results

### 3.1. Textile Capacitive Sensor Performance

Figure 10 depicts the textile capacitive sensor trend obtained from performance testing under the various forces applied. The textile capacitive sensor trend behaved linearly and non-linearly when the applied pressure was below and above 11.00 kPa, respectively. The sensitivity of our textile capacitive sensor was 0.232 MPa^−1^ and 0.070 MPa^−1^ for using pressures lower and higher than 11.0 kPa, respectively.

### 3.2. Data after Processing

Figure 11 demonstrates the result of applying the Zero-To-Zero detection algorithm to mid-1-min foot plantar pressure data. Figure 12 presents the result of normalizing a GC that provides 100 data points and then reconstructs the gait event from toe-off to heel strike. During the experiment, all participants were allowed to self-select their walking speed. We obtained an average walking speed of 2.40 ± 0.30 km/h for comfortable gait and 5.10 ± 0.20 km/h for fast gait. Figure 13 shows the result of input data and reference data after processing a GC from a random subject.

### 3.3. ANN Models Evaluation

The summary result for RMSE, MRE, and R to evaluate each model’s accuracy on GC data is illustrated in Table 3, Table 4, and Table 5. They show the efficiency of five models for predicting the knee joint angle for both comfortable and fast walking speeds by shoe type: S1 (flat), S2 (sneaker), and S3 (stiletto). The implementation of all ANN models to predict the knee joint angle when wearing S1, S2, S3 walking at comfortable and fast gaits showed a (minimum, maximum) of (5.60, 21.32), (0.03, 0.12), (0.75, 0.97) for RMSE (deg), MRE, and R-value, respectively. The implementation M6 to predict the knee joint angle when wearing S1 and S2 walking at comfortable and fast gaits showed an R-value was higher than 0.90 with MRE lower than 15%. The implementation M6 to predict the knee joint angle when wearing S3 showed an R-value of 0.90 and 0.82 for comfortable and fast gait, respectively. 

Table 6 illustrates the ANN and CNN approach’s result under considering the shoe base in predicting knee joint angle. Using S1, five ANN models provide R-value higher than 0.90, but M2 provides 0.88 of R-value. Using S2, five ANN models provide R-value higher than 0.90, but M3 provides 0.81 of an R-value. However, using S3, all ANN models give an R-value lower than 0.90. For CNN approach, all CNN model provides R-value higher than 0.90 when wearing S1 and S2. Using S3, five ANN models provide R-value lower than 0.90, but M3 yields an R-value of 0.90. 

Table 7 shows quantifiable measures; mean bias and limits of the agreement give information about the utility of each ANN model in estimating knee joint angle compare with actual value. The mean bias is represented by the gap between the *X*-axis, regarding the zero differences between estimating using each ANN model and actual knee joint angle as regression analysis. In our data set, the Upper LOA (Limit of Agreement) can be calculated using mean bias + 1.96 SD (Standard Deviation), and the Lower LOA can be calculated using mean bias − 1.96 SD. Table 7 illustrates the positive mean bias when comparing the knee joint angle estimate of M1, M5, and M6 with actual values. And the negative mean bias was found with M2, M3, and M4 compare to actual value. Figure 14 indicates the Bland-Altman plot showing the difference between estimation result and actual knee joint angle for three shoe types and two gait speeds regarding each ANN model. 

### 3.4. Knee Flexion Pattern

Figure 15 shows the knee flexion patterns during a GC by the three shoe types while walking at a comfortable gait and a fast gait. ANN deep learning predicted these well, and provided similar patterns compared to actual patterns.

## 4. Discussion

This paper presents the development of a textile capacitive sensor that has two dielectric layers, which are placed between three electrodes to measure foot plantar pressure. Our findings indicated that when applying pressure below and above 11.00 kPa, our sensor’s sensitivity was 0.232 MPa^−1^ and 0.070 MPa^−1^, respectively. According to the studies by Zhang et al., and Duc et al., implemented pressure at different ranges varies the sensitivity of the sensor, and the thinner the dielectric film, the higher the sensor sensitivity [28,36]. Therefore, the other pressure ranges and the dielectric thickness might affect our sensor’s sensitivity.

Our study also investigated the performance of ANN based on foot plantar pressure to estimate knee joint angle by three shoe types in subjects who walked at comfortable and fast speeds. Previous studies stated that if the correlation coefficient between the predicted and the actual measure is higher than 0.90, and the MRE is less than 15%, then the prediction model is reliable and valid [23,37]. Moreover, the R-value lay between 0.70 to 0.90 is also empathized a strong correlation [38] between foot plantar pressure and knee joint angle estimation. After comparing ANN and CNN approaches, it could be found that correlation coefficients of the knee joint prediction are similar regarding three shoe types. 

In our work, the estimated results for knee joint angle using foot plantar pressure of three testing shoes provided a low RMSE, great model fitting, and strong correlation when implementing M4 and M5 for comfortable gait and fast gait speed, respectively. So, the obtained accuracies were higher when using speed-based ANN models than shoe-based ANN models. Table 1 presents the relatively small sample size used in shoe-based ANN models compared to speed-based ANN models that may have affected model training and interpretation of this result.

We found a similar result compared to the previous research, which integrated a deep learning approach and foot plantar pressure to estimate lower limb joint angles. Sivakumar et al. (2018) estimated the ankle joint angle from ground reaction forces and a Feed Forward Neural Network deep learning approach. They found a high Pearson correlation coefficient (ρ > 0.94) when estimating ankle joint angle during the stance phase [30]. Xie et al., (2020) predicted the lower limb joint angle base on multi-source signals using the general regression neural network optimized by golden section algorithm for subjects wearing an exoskeleton [23]. They employed a combination of three different types of data sources; hip angle electromyography, and plantar pressure to estimate the lower limb hip, knee, and ankle joint angles. From their study, the knee joint angle estimation had a high correlation coefficient of 0.99. Their algorithm provides a high accuracy, which may be due to the combination of multi-source data, which could improve deep learning models during training [23]. 

The Bland-Altman plot analysis is a method to evaluate a bias between the mean differences and to estimate an agreement interval, within which 95% of the differences of knee joint angle estimate regarding each ANN model and actual data. Our finding indicated that few scatter points move off the limit lines (Lower LOA and Upper LOA), and some scatter points lie close to the mean bias line (Figure 14). A good agreement between the two methods is shown through the scattering points lie relatively close to the mean bias line [39]. We found the mean bias is not zero, which indicates the systematic error between the estimate and actual knee joint angle. In this study, each ANN model was built with a relatively small sample size with no pressure data appear during the swing phase, yielding a lack of information for training the model. This result may indicate that the sample size and no foot pressure data during the swing phase affect model training, influencing agreement between the estimate and actual knee joint angle.

Our study revealed that the knee flexion pattern was similar to actual data when applying ANN deep learning. These results indicate that our textile capacitive sensor performs well and indirectly detects knee flexion angle across gait periods during two gait speeds for three different shoe types, even with high heels. During the swing phase, the large amplitude difference indicates that it was more challenging to predict knee joint angle due to a lack of signal because of the foot plantar pressure commonly present during the stance phase. This makes it difficult for ANN deep learning algorithms to learn. Our study is applicable for predicting joint angles for the whole GC; even though the plantar pressure information was only present during the stance phase. Typically, gait disorders or abnormalities could happen at any stage of the GC. Example, wearing high heels decreased knee flexion during the swing phase [40], resulting in a stiff-knee gait that may present as a gait abnormity. It is essential to measure the joint angles for the whole GC for further analysis. Therefore, this approach might be useful for outdoor monitoring on joint angle performance during the GC, especially for high heel wearers with different gait speeds.

This study has limitations, which subsequent studies will overcome. First, our textile capacitive sensor only measured normal stress during walking; while walking with a high heel, the stress may be subject to normal stress and shear stress. So, our sensor may not have acquired all data for the ANN model to yield high accuracy for S3 estimation. Second, our work limited the number of sensors in each insole with ten; increase the number of sensors set up in the insole may induce less space between each sensor. So, less lacking foot plantar pressure measurement provides enough information for training deep learning models to achieve a highly accurate estimation. Third, our ANN models were used young subject data and a relatively small sample size; even 30 GCs may significantly contribute to the ANN capacity to learn during training, resulting in an MRE of less than 15%. Typically, a generalized deep learning model requires a large dataset because the deep learning approach can train using direct input features without employing a feature extraction. Our study was under pilot test, but our finding might contribute necessary information for the leading research, which should consider the primary variable (shoe type and gait speed) that would affect model training when using foot pressure data. Fourth, our ANN models were built with a dataset that has a small range of shoe height and gait speed. We only investigated the capability of the development sensor to measure foot plantar pressure on three shoe types with heights of 0.01 m, 0.03 m, and 0.09 m. Moreover, we did not consider a wide range of walking speeds which means we should add more categories of walking speeds, i.e., slow and medium speeds, that can identify which walking speed is best for building ANN models. The foot plantar pressure may vary depending on shoe height and gait speed [41]. Large range datasets could be made model well training that can generalize quickly.

Although our work has limitations, based on our preliminary investigation, we found that the foot plantar pressure measure through the textile capacitive sensor could integrate with the ANN approach to estimate knee joint angle. Moreover, we also found that shoe type and gait speed play an essential role, which should be considered before creating an ANN model using foot plantar pressure to estimate knee joint angle. Future studies should investigate those issues to achieve a generalized model that can apply with various shoe types and gait speeds for estimating knee joint angles. 

## 5. Conclusions

The development of a textile capacitive sensor with a less complex structure resulting in low-cost operation can measure the foot plantar pressure even for high-heel wearers walk at comfortable and fast speed. The integration of ANN and foot plantar pressure data obtained from three shoe types with two walking speeds could provide more accurate data to make functional models predict knee joint angles. We found all five ANN models provided a high correlation efficiency of 0.75 to 0.97 for three testing shoes with two gait speeds. Comparing shoe type and gait speed variables, speed-based ANN yields higher efficiency in estimating knee joint angle. Moreover, analysis through Bland-Altman also shows an agreement between actual and estimate knee joint angle through the deep learning approach using foot pressure data. Our research supports evidence that the ANN model and foot plantar pressure provide a strong correlation that predicts knee joint angle for a GC. This method may provide a potential opportunity for outdoor gait monitoring to follow up knee joint angle performance during walking, especially for high heel wearers. 

## Figures and Tables

**Figure 1 sensors-21-05484-f001:**
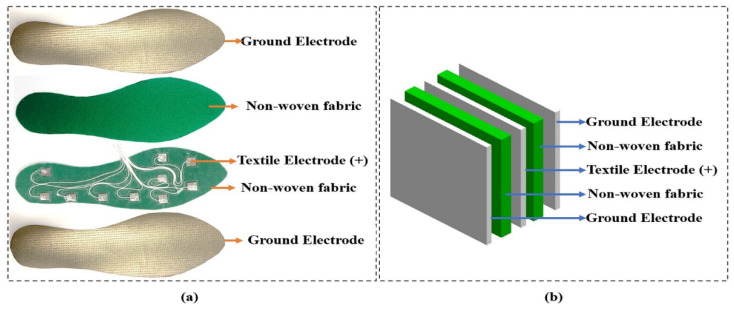
(**a**) The design structure for textile capacitive sensor; (**b**) equivalence structure of each layer.

**Figure 2 sensors-21-05484-f002:**
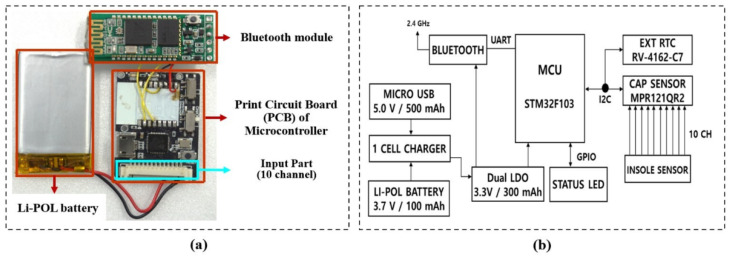
(**a**) schematic of hardware design to measure the capacitance of the textile capacitive sensor; (**b**) detail and connection of hardware design [32].

**Figure 3 sensors-21-05484-f003:**
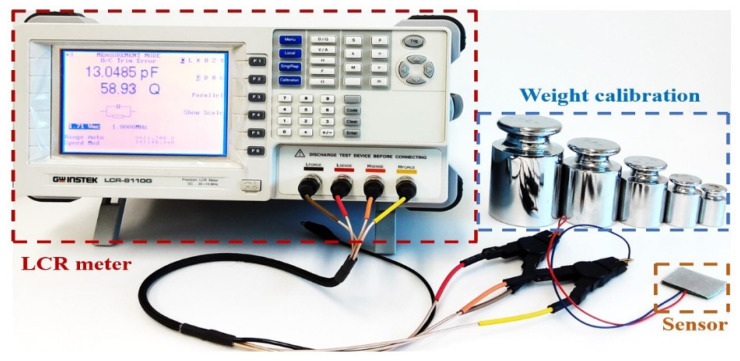
Set up for testing textile capacitive sensor performance.

**Figure 4 sensors-21-05484-f004:**
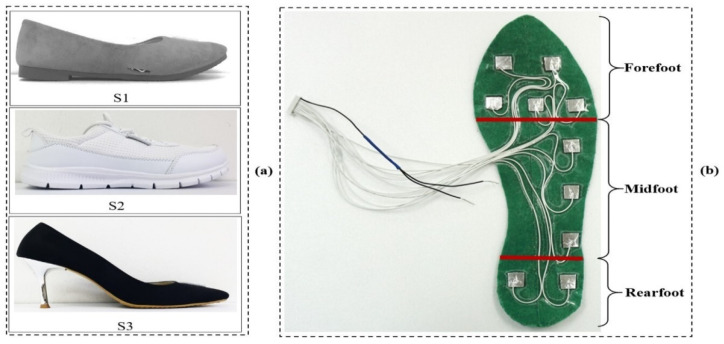
(**a**) Lateral view of the three shoe types: flat shoe (S1), sneaker (S2), and stiletto heel (S3); (**b**) The sensor set up on the insole is based on three main foot areas: forefoot, midfoot, and rearfoot.

**Figure 5 sensors-21-05484-f005:**
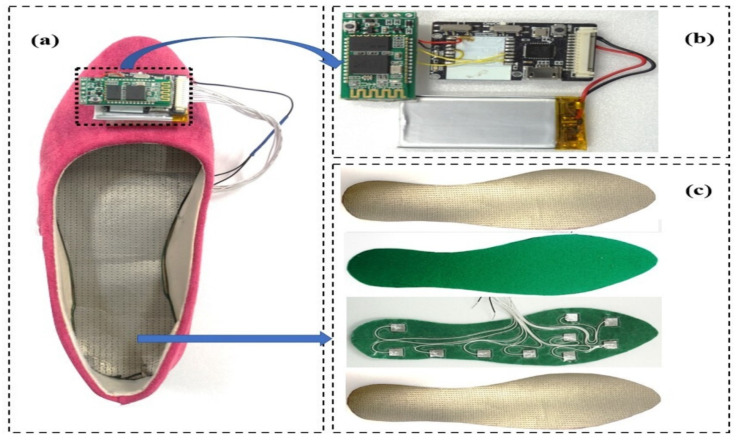
(**a**) The whole system setup to measure foot pressure; (**b**) Hardware design to receive and transmit data; (**c**) Sensor layers which mount in the shoe.

**Figure 6 sensors-21-05484-f006:**
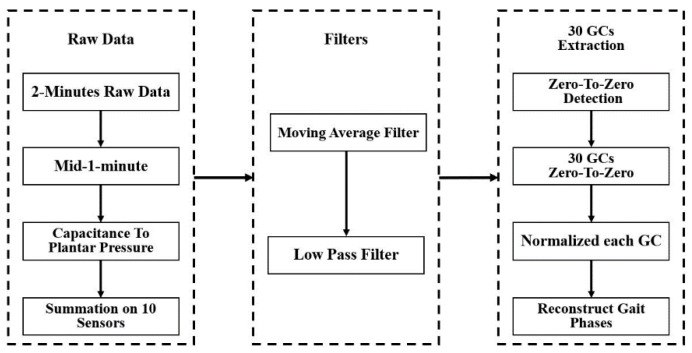
The process of extracting 30 gait cycles of foot plantar pressure that each gait cycle contains 100 data points.

**Figure 7 sensors-21-05484-f007:**
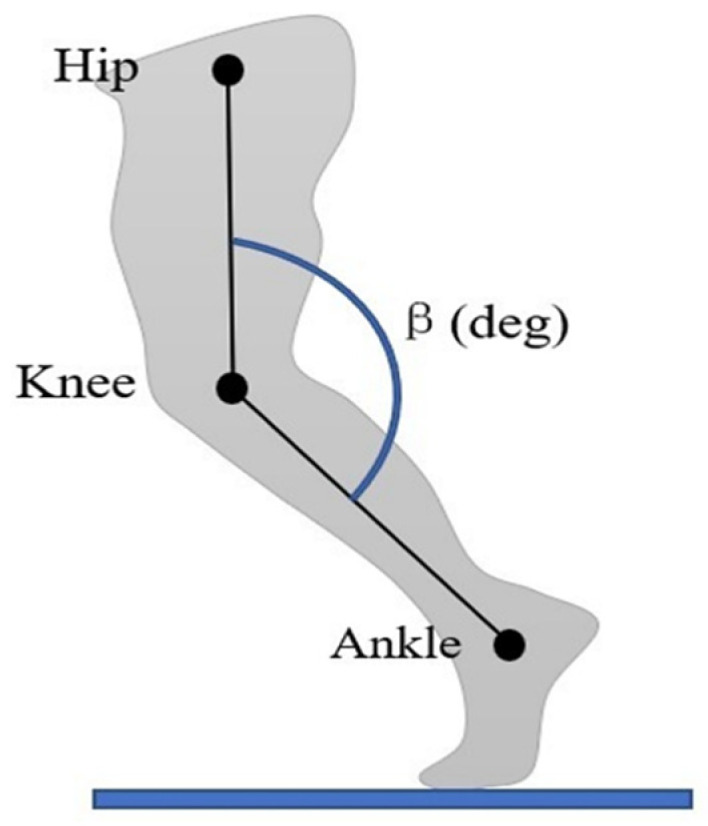
The technique of calibration knee joint angle (β) in Kinovea software.

**Figure 8 sensors-21-05484-f008:**
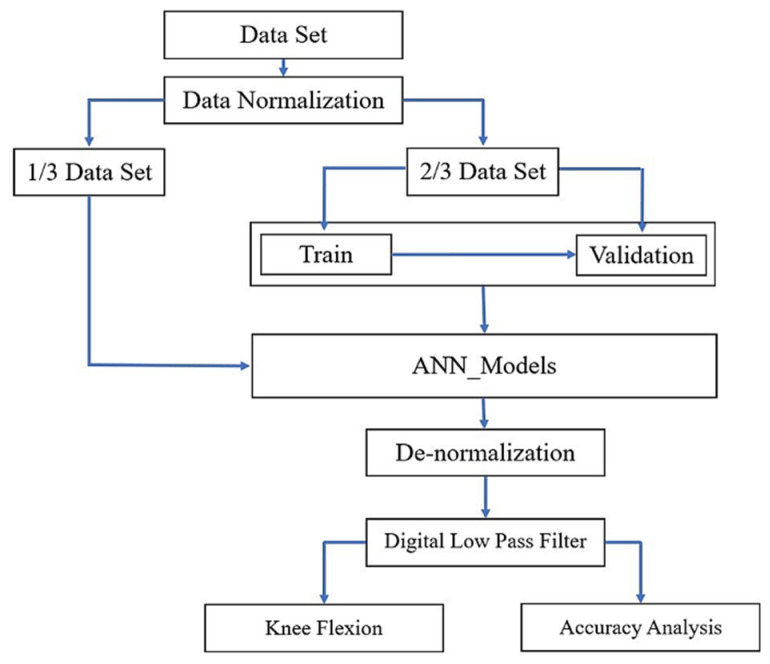
The whole workflow of ANN models for prediction the knee joint angle.

**Figure 9 sensors-21-05484-f009:**
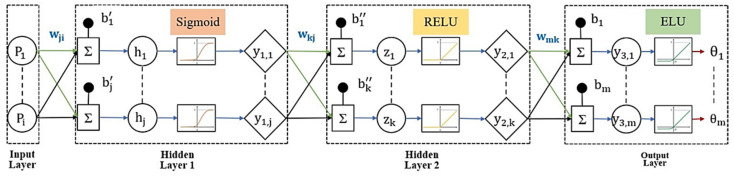
Block diagram of the four layers ANN model.

**Figure 10 sensors-21-05484-f010:**
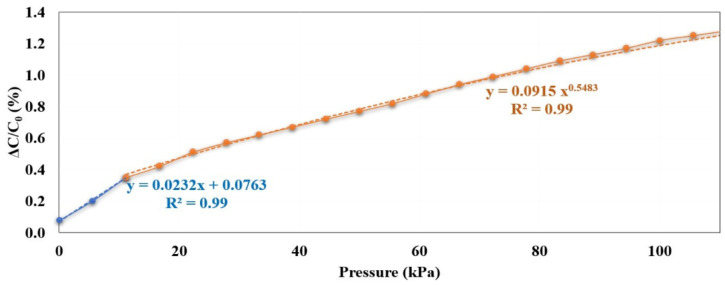
Result on changing capacitance versus applying pressure.

**Figure 11 sensors-21-05484-f011:**
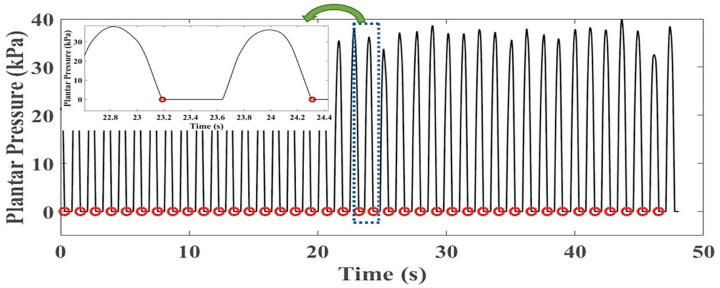
A sample result on Zero-To-Zero detection to mid-1-min foot plantar pressure.

**Figure 12 sensors-21-05484-f012:**
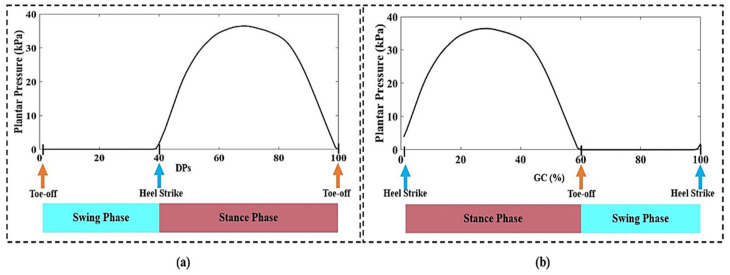
(**a**) Result of 100 data points (DPs) of foot plantar pressure in a gait cycle that start at toe-off to toe-off gait event; (**b**) A gait cycle (%) of foot plantar pressure that start at heel strike to heel strike gait event.

**Figure 13 sensors-21-05484-f013:**
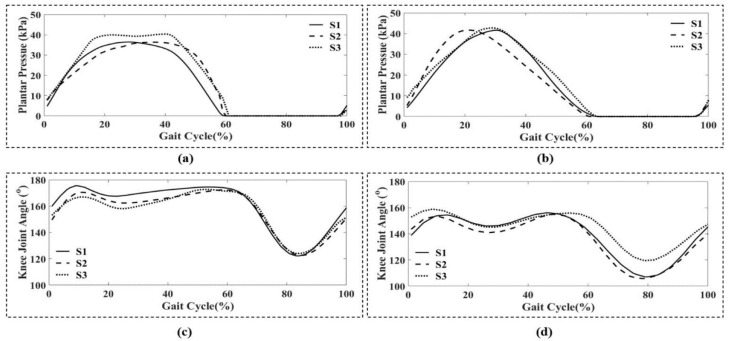
A sample of input data versus reference data in a gait cycle for three testing shoes, S1 (flat), S2 (sneaker), S3 (stiletto heel), was used to create the ANN model. (**a**) Foot plantar pressure at comfortable gait; (**b**) Foot plantar pressure at fast gait; (**c**) Knee joint angle at comfortable gait; (**d**) Knee joint angle at fast walking.

**Figure 14 sensors-21-05484-f014:**
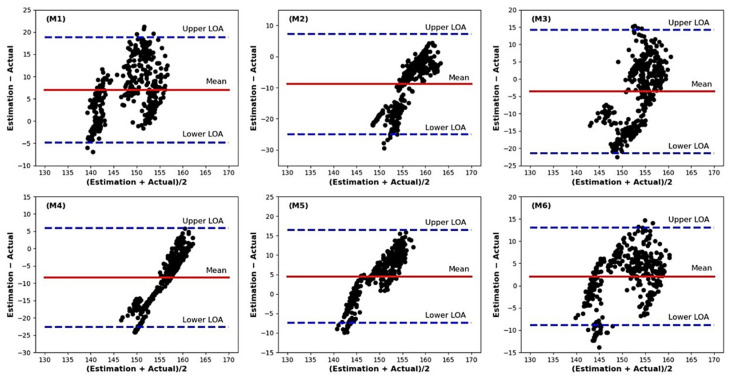
The Bland-Altman plot showing the difference between estimation result and actual knee joint angle for all shoe types and two gait speeds regarding to each ANN model, M1 (S1 model), M2 (S2 model), M3 (S3 model), M4 (comfortable gait model), M5 (fast gait model), M6 (model using all condition of dataset).

**Figure 15 sensors-21-05484-f015:**
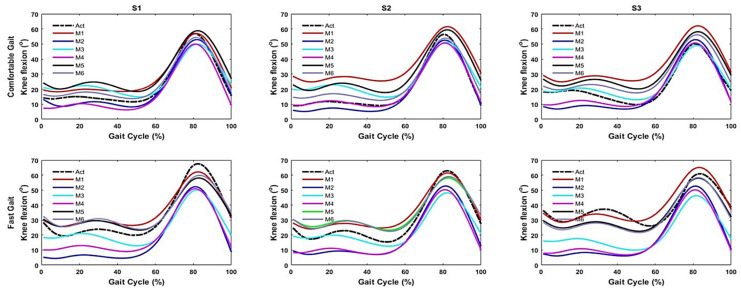
Result of knee flexion pattern of the gait cycle of six models with actual data for three different type of testing shoes (columns) at two gait speeds (rows); S1 (flat), S2 (sneaker), S3 (stiletto heel), Act (actual knee joint angle), M1 (S1 model), M2 (S2 model), M3 (S3 model), M4 (comfortable gait model), M5 (fast gait model), M6 (model using all conditions of the dataset).

**Table 1 sensors-21-05484-t001:** The characteristics of three testing shoes.

Shoe Characteristics	S1	S2	S3
Heel height (cm)	1.0	3.0	9.0
Ball width (cm)	8.0	9.5	8.0
Shoe size (cm)	23.5	23.5	23.5
Heel base (cm^2^)	28.5	55.3	0.9

S1: flat shoe; S2: sneaker, S3: stiletto heel.

**Table 2 sensors-21-05484-t002:** The condition of the data set used to form six ANN models.

Shoe Type	Speed	Model Name	Number of Gait Cycle for Training ANN Model
S1	C and F	M1	300
S2	C and F	M2	300
S3	C and F	M3	300
S1, S2, S3	C	M4	450
S1, S2, S3	F	M5	450
S1, S2, S3	C and F	M6	900

S1 (flat), S2 (sneaker), S3 (stiletto heel), C (comfortable speed), F (fast speed).

**Table 3 sensors-21-05484-t003:** Summary of the accuracy analysis of each model with S1 (flat) testing data.

Models	Comfortable Gait			Fast Gait		
	RMSE	MRE	R	RMSE	MRE	R
M1	7.10	0.04	0.95	8.18	0.05	0.94
M2	5.08	0.03	0.96	18.90	0.10	0.81
M3	7.50	0.04	0.91	11.46	0.06	0.94
M4	6.46	0.03	0.96	14.52	0.08	0.91
M5	9.11	0.05	0.94	8.79	0.05	0.92
M6	6.60	0.04	0.91	8.61	0.05	0.92

S1 (flat), S2 (sneaker), S3 (stiletto heel), M1 (S1 model), M2 (S2 model), M3 (S3 model), M4 (comfortable gait model), M5 (fast gait model), M6 (model using all condition of dataset).

**Table 4 sensors-21-05484-t004:** Summary of the accuracy analysis of each model with S2 (sneaker) testing data.

Models	Comfortable Gait			Fast Gait		
	RMSE	MRE	R	RMSE	MRE	R
M1	15.56	0.10	0.94	7.45	0.04	0.94
M2	5.53	0.03	0.96	13.21	0.07	0.87
M3	9.92	0.05	0.86	12.50	0.07	0.75
M4	4.93	0.03	0.96	12.54	0.07	0.92
M5	10.78	0.06	0.95	7.73	0.04	0.96
M6	6.10	0.03	0.95	6.90	0.04	0.97

S1 (flat), S2 (sneaker), S3 (stiletto heel), M1 (S1 model), M2 (S2 model), M3 (S3 model), M4 (comfortable gait model), M5 (fast gait model), M6 (model using all condition of dataset).

**Table 5 sensors-21-05484-t005:** Summary of the accuracy analysis of each model with S3 (stiletto heel shoes) testing data.

Models	Comfortable Gait			Fast Gait		
	RMSE	MRE	R	RMSE	MRE	R
M1	14.13	0.09	0.90	6.69	0.04	0.85
M2	7.63	0.04	0.91	21.32	0.12	0.81
M3	5.60	0.03	0.91	17.55	0.10	0.85
M4	6.55	0.03	0.90	20.92	0.12	0.79
M5	10.55	0.06	0.92	7.23	0.04	0.87
M6	8.46	0.05	0.90	8.67	0.05	0.82

S1 (flat), S2 (sneaker), S3 (stiletto heel), M1 (S1 model), M2 (S2 model), M3 (S3 model), M4 (comfortable gait model), M5 (fast gait model), M6 (model using all condition of dataset).

**Table 6 sensors-21-05484-t006:** Comparison of ANN and CNN approach for each shoe types.

Models	ANN			CNN		
	S1	S2	S3	S1	S2	S3
M1	0.94	0.94	0.88	0.93	0.94	0.87
M2	0.88	0.91	0.86	0.92	0.94	0.88
M3	0.92	0.81	0.88	0.92	0.93	0.90
M4	0.94	0.94	0.85	0.92	0.93	0.83
M5	0.93	0.95	0.89	0.93	0.94	0.88
M6	0.92	0.96	0.86	0.94	0.95	0.88

S1 (flat), S2 (sneaker), S3 (stiletto heel), M1 (S1 model), M2 (S2 model), M3 (S3 model), M4 (comfortable gait model), M5 (fast gait model), M6 (model using all condition of dataset).

**Table 7 sensors-21-05484-t007:** Summary of mean bias, Upper LOA, Lower LOA obtained from the Bland-Altman.

	M1 & Act	M2 & Act	M3 & Act	M4 & Act	M5 & Act	M6 & Act
Mean bias	7.01	−8.85	−3.58	−8.36	4.53	2.05
Upper LOA	18.87	7.24	14.25	5.95	16.49	13.00
Lower LOA	−4.84	−24.95	−21.41	−22.68	−7.44	−8.89

Six ANN models (M1, M2, M3, M4, M5, and M6), Act (Actual knee joint angle).

## Data Availability

All available processing data and code required to create the deep learning models are available from GitHub at the following address https://github.com/Vantha17/ProcessingData_Code.git accessible as of 13 August 2021.

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
