# Peer review of "Estimation of Knee Joint Angle Using Textile Capacitive Sensor and Artificial Neural Network Implementing with Three Shoe Types at Two Gait Speeds: A Preliminary Investigation"

_sensors, 2021, doi:10.3390/s21165484_

Round 1

Reviewer 1 Report

Major/Minor comments:

The manuscript is suffering with lack of experimental work or evidence, where author should focus more in revised manuscript.

  1. Page 2, Figure: Why author used the complex structure of the textile capacitive sensors using 3 electrodes and 2 dielectric materials (nonwoven) instead of 2 electrode and 1 dielectric (nonwoven)?
    1. Author should show the real demonstration/picture of the sensor
    2. As a dielectric material textile nonwoven is always uneven (surface) by nature, which should show very inconsistent in capacitive response by pressure. Such complex structure based on more electrodes (3) and dielectric material (2) should enhance the instability of the response.
  2. Capacitive sensors are prone to hysteresis, this effect was not discussed.
  3. Page3: Line 127-134: Author discussed about the sensitivity of the capacitive sensor by the thickness, which is not really optimized in this work?
  4. Page 4 in Figure 2 and page 10 Figure 8: Author showed that a calibration curve of capacitance changes vs pressure using different loads (metallic weight by 0.5 kg differences). The interference between the capacitive sensor and metallic conductor (load) has not been studied, as we know the capacitance changes itself, while a conductive (load) surface come into contact with the sensors. How author considered this issue while characterization?
  5. Page 10, Figure 8: Author has calibrated the sensor within the load 0-10 Kg (Page 4, line 142, while in application using shoe is considering weights over 50 kgs! Where the capacitance change is not linear particularly at higher load... If so, how Author will validate the results?
  6. Page 4, figure 3 (b): are they 3 matrices of capacitive sensors in 3 areas? The whole sensor design/electronics/connection should be more informative and clearly illustrated for the reader. A real picture of the insole matrix containing 10 sensors should be presented in the picture
  7. Page 6, Figure 4: How the hardware is implemented in the system? How the Algorithm is used to realize sensor data in foot planter pressure?
  8. Page 6, figure 4: line 192-202: How the data was collected?
  9. Page 8: Architecture of the ANN model should be explained.
  10. Page 8, Table 2; page 12, table 3 and 4: The representation of the data or results very confusing e.g, Fast gait and MF or comfortable gait and MC etc..

Author Response

Dear Reviewer, 

We sincerely appreciate constructive comments from the reviewers. Moreover, all revise parts have been highlighted in red color with marked up the “Track Changes” in Revise Manuscript. We have addressed all of the review comments carefully and revised the manuscript based on them.

Best regards,

Chhoeum Vantha

Reviewer 2 Report

The manuscript by Chhoeum et al. details a method developed to estimate knee joint angles using a textile capacitive sensor and an artificial neural network model. Overall, the manuscript is clear and well presented, and the authors’ reasoning was generally easy to follow (although see below). While the paper is interesting, this reviewer feels there are 5 key points that need to be addressed before it is potentially suitable for publication.

1) Significance/Impact

The significance/impact is not immediately clear to me. For example; Line 39: “It is essential to realize the effect of gait speed and shoe type on the knee joint angle, which can lead to early diagnosis and treatment to prevent pain and injuries.” This statement is not particularly convincing especially since there are no citations to back it up. Also, as the study only focused on women, do these findings also extend to men?

I think the authors really need to point out the potential application(s) of this new approach. An important one that immediately comes to mind may be for freezing of gait in Parkinson’s disease (e.g., this may be an easier approach than Fig. 9 in https://www.ncbi.nlm.nih.gov/pmc/articles/PMC6567147/pdf/sensors-19-02416.pdf).

2) I get the spin for ANN versus CNN (lines: 93-109). But what about other types of models (e.g., SVM regression)?

3) I am not sure the “R-value” metric used here is the most appropriate measure. Plotting the predicted values versus the actual values would help the reader visualize the data and would be quite informative (see below). The authors may want to also consider a Bland-Altman plot/analysis, or provide some justification as to why they chose this particular “R-value” metric.

Also, the discussion related to accuracy and correlation coefficients (e.g., Line 429: “high correlation coefficient”) should be made with caution as these are not necessary capturing the same thing. For instance, consider the variables X={1,2,3,4,5} and Y={0.1,0.2,0.3,0.4,0.5}. Here, the correlation coefficient would be equal to 1 (perfect correlation), yet the actual accuracy is an order of magnitude off.

4) There doesn’t appear to be an ethics statement.

5) Additional English correction is needed (a few examples below):

e.g., line 324-325: “xi is the perdition,…”

         line 190: “had 5-mine breaks in between.”     

Author Response

(The authors gave the same response as above.)

Reviewer 3 Report

In general, the organization and writing are fine. However, the novelty is limited while there are several technical issues in the methods and experiments. The major comments are listed as follows:

  1. Seven subjects are too few to support the reliability of the proposed approach while using artificial neural networks.
  2. Only “young” subjects are recruited for the experiments. The subjects with different weights, BMI, and age should be involved to validate the feasibility of the systems.
  3. Other DL models including CNN and RNN should be applied to compare with the proposed approach. These models are also state-of-the-art models in ANN.
  4. Please use eave-one-subject-out cross-validation approaches to validate the reliability of the proposed approach.
  5. Currently, five ANN models are trained for different shoe sizes and speeds. Please test and show the results using a general model for all conditions. It can help readers understand the feasibility (or challenges) of the proposed system.

Author Response

(The authors gave the same response as above.)

Reviewer 4 Report

Interesting paper. However I have some suggestion.

  • The measurement frequency for sensor measurement should be indicated and also the capacitance measurement range.
  • The authors are using Phython, MATLAB, EXCEL, Kinovea. It should explain the reason to use all theses software.
  • A real image of the developed devices will be apreciated. In order to see the real connection and location of the schematic shows on Fig.1b.
  • Seven young are used? It is enough? In my experience this number is too low. This decision should be justified.

Author Response

(The authors gave the same response as above.)

Round 2

Reviewer 2 Report

1) If human subjects are used, isn't an ethics statement required?

2) The systematic and possible relative bias that appears in the Bland-Altman plot/analysis should be discussed.

Author Response

Dear Reviewer,

Thank you very much for spending your time providing us a great comment and pointing out a missing point. We have addressed the reviewer’s comments on the attached field of RESPONSE_ REVIEWER 2_Round 2. And the detail of the explanation has been added to Revised Manuscript Round 2.

Have a nice day!

Best regards,

Chhoeum Vantha

Reviewer 3 Report

The authors have tackled the reviewer's comments. Currently, the reviewer suggests accepting the paper.

Author Response

Dear Reviewer,

We thank you for let us know the good news. Moreover, thank you very much for spending your time providing us a great comment and pointing out a missing point to improve the quality of our paper.

Have a nice day!

Best regards,

Chhoeum Vantha